# ReSyn: Autonomously Scaling Synthetic Environments for Reasoning Models

## Abstract

Reinforcement learning with verifiable rewards (RLVR) has emerged as a promising approach for training reasoning language models (RLMs) by leveraging supervision from verifiers. Although verifier implementation is easier than solution annotation for many tasks, existing synthetic data generation methods remain largely solution-centric, while verifier-based methods rely on a few hand-crafted procedural environments. In this work, we scale RLVR by introducing ReSyn, a pipeline that generates diverse reasoning environments equipped with instance generators and verifiers, covering tasks such as constraint satisfaction, algorithmic puzzles, and spatial reasoning. A Qwen2.5-7B-Instruct model trained with RL on ReSyn data achieves consistent gains across reasoning benchmarks and out-of-domain math benchmarks, including a 27% relative improvement on the challenging BBEH benchmark. Ablations show that verifier-based supervision and increased task diversity both contribute significantly, providing empirical evidence that generating reasoning environments at scale can enhance general reasoning abilities in RLMs.

## 1 Introduction

Pioneered by OpenAI-o1 and DeepSeek-R1 (OpenAI, 2024; DeepSeek-AI, 2025), recent work has demonstrated that reinforcement learning (RL) can substantially enhance the reasoning capabilities of large language models (LLMs) by training on math and coding data. These RL-trained models exhibit "emergent behaviors" – including backtracking, self-verification, and reasoning over long chains of thought – that correlate strongly with task success (Gandhi et al., 2025). A key insight from this line of research is that such behaviors do not need to be imitated from human demonstrations; instead, they can be surfaced by reinforcing reasoning-intensive outcomes, such as arriving at the correct solution to a math problem.

This line of work has largely focused on math and code, where correctness can be checked against reference answers or unit tests. While this provides a clear reward signal, it also constrains us to problems with known ground-truth outputs, limiting both the quantity and difficulty of problems available for RL. By contrast, many other reasoning tasks encourage similar problem-solving behaviors while being much easier to verify. For instance, completing a Sudoku puzzle can be challenging, but verifying a filled grid is straightforward. Likewise, tasks such as dependency sorting and constraint satisfaction require long reasoning chains but can be verified with simple rule checks. Even when efficient algorithms exist, reasoning about these tasks in natural language challenges LLMs to plan multi-step strategies, track intermediate states, and correctly apply symbolic rules.

Several studies have begun to explore code-based, puzzle-like reasoning environments for LLM training (Pan et al., 2025; Liu et al., 2025a; Stojanovski et al., 2025). These works show that, despite their simplicity, training in these environments can improve downstream performance on reasoning benchmarks and elicit behaviors similar to those observed in math training. However, prior efforts have relied on a small set of manually curated environments, where problem instances are procedurally generated according to hand-crafted logic. While this strategy is effective at generating a lot of data, it offers limited variation in reasoning patterns, producing repetitive problems that may fail to elicit more generalizable skills. Ultimately, task diversity is limited by the manual effort required to design new environments.

Figure 1: Overview of synthetic environment generation in the ReSyn data pipeline. An LLM is prompted with seed keywords to synthesize Python implementations of reasoning environments, each defining instance generation $\rho_0$, observation $O$, and reward $R$. The generated environment is evaluated by an LLM judge. Ones that pass are added to the ReSyn dataset, while failed ones are revised with feedback and re-evaluated.

In this paper, we scale up this approach by automatically generating a diverse collection of reasoning environments. Each synthetic environment is equipped with a problem generator and a verifier, both implemented in LLM-synthesized code. This design combines the diversity of model-based data generation with the scalability of procedural instance generation. Ablation experiments show that scaling the number of environments, controlled for the total number of instances, leads to substantial improvements in downstream performance. This suggests a promising avenue for automatically improving LLM reasoning capability without requiring manual effort to create new tasks.

Compared to prior work on LLM-based synthetic data generation for RL (Guo et al., 2025; Havrilla et al., 2025), which relies on model-generated reasoning as ground truth, our pipeline synthesizes code-based verifiers to provide rewards during RL. This enables reliable supervision for problems that exceed the natural language reasoning capacity of the teacher model, either by leveraging computational tools or by exploiting the fact that many tasks are easier to verify than to solve. Our ablations confirm that utilizing verifiers grounded in code provides better supervision than training on model-generated solutions, resulting in a 14% relative improvement (vs. 4% with no code or verifier) from the original -Instruct model on BBH.

Using this pipeline–termed RESYN–we expand the scale and diversity of synthetic reasoning environments by over an order of magnitude compared to previous work. Training models with RL on RESYN data leads to substantial gains on general reasoning and math benchmarks (+14% and +27% relative improvement on BBH and BBEH, respectively), demonstrating that diverse, verifier-driven synthetic tasks form an effective foundation for developing stronger reasoning models.

## 2 RESYN DATA PIPELINE

Typical reasoning datasets consist of pairs of questions and reference answer $(Q, A)$, where model outputs are judged by matching against $A$. This setup is limiting: many reasoning problems admit multiple valid solutions, and it is often more natural to specify how to *verify* correctness than to provide a single reference answer. Motivated by this, we represent each problem as a pair $(Q, V)$, where $V$ is a verifier that maps a candidate solution to a binary reward. Our goal is to generate a diverse dataset of such pairs using an LLM.

While LLMs can directly generate question–answer pairs for reasoning (Guo et al., 2025), procedural generation methods like that in Liu et al. (2025a); Stojanovski et al. (2025) offer other important advantages. First, they allow virtually unlimited data to be generated by executing a simple program, often with controllable difficulty. Second, a single verifier can be reused across many instances, avoiding the cost of creating a new one for each problem.

Shown in Fig. 1, our pipeline combines these benefits by leveraging the coding capabilities of LLMs. Rather than creating individual $(Q, A)$ pairs, it generates synthetic environments $E$, implemented in code, that encapsulate both instance generation and verification logic. The final $(Q, V)$ pairs used for RLVR training are then obtained by sampling from the state distribution of these environments.

## 2.1 PROBLEM FORMULATION

Let $\Sigma^*$ denote the set of all strings, including all natural language questions and potential responses from an LLM. We model a reasoning environment as a interactive learning problem $\mathcal{T} = (\mathcal{S}, \mathcal{A}, R, O, \rho_0)$, where $\mathcal{S}$ is a set of problem instances, where $s \in S$ represents an individual problem example. We define $\mathcal{A} = \Sigma^*$ is the action space – natural language responses from the LLM, which we regard as a policy.

- $\mathcal{S}$ is the space of instance parameters, where each $s \in \mathcal{S}$ specifies a different problem instance.
- $\mathcal{A} = \Sigma^*$ is the action space – natural language responses from the policy LLM.
- $O : \mathcal{S} \to \Sigma^*$ is the observation function, mapping from instance parameters to natural language questions that can be understood by a human or LLM.
- $R : \mathcal{S} \times \mathcal{A} \to \{0, 1\}$ is the reward function, checking whether a response $a \in \mathcal{A}$ is a correct solution to problem instance $s \in \mathcal{S}$.
- $\rho_0(d) \in \Delta(S)$ is the initial state distribution over $S$, which defines the distribution of problem instances. We parameterize it with an adjustable difficulty level $d$.

## 2.2 DATA PIPELINE

Our synthetic data generation pipeline starts from a small set of seed topics and produces a large collection of $(Q, V)$ pairs. It comprises five stages:

**1. Keyword/Topic Extraction.**

We first compile a list of topic keywords that span diverse reasoning skills. To do this, we draw on two sources. (1) We show an LLM one problem from each subtask of Big-Bench Hard and KOR-Bench (Ma et al., 2024), instructing it to propose several relevant keywords; after deduplication this produces roughly 100 candidates. All keywords are 1-2 word phrases, making data leakage unlikely. (2) We manually augment this set with algorithm- and data-structure–related keywords, which empirically yield high-quality, verifiable reasoning tasks. We also filter out topics ill-suited for rule-based verification. While we use BBH and KOR-Bench as seed sources, their specific choice is not essential–they simply offer a reasonably diverse set of problem categories. We provide the final keyword list in Appendix A.1.

**2. Task Synthesis.** For each keyword, we prompt the LLM to come up with a related reasoning task and implement it in Python with instructions for a specific code structure. We prompt the LLM $n = 8$ times for each keyword to generate multiple related environments. The LLM is instructed to implement functions defining the state distribution $\rho_0$, observation function $O$, and reward function $R$:

- $R(s, a)$ is implemented as a method with signature $\mathcal{S} \times \mathcal{A} \to \{0, 1\}$. In practice, this method extracts the candidate answer from a response $a \in \mathcal{A}$, executes problem-specific verification against instance parameters $s \in \mathcal{S}$, and returns 1 if correct.
- $O(s)$ is a method that creates a natural language question from instance parameters $s$. It usually does this by inserting instance parameters into a string template. The LLM is instructed to ensure that the natural language question is well-specified.
- $\rho_0(d, n)$ is a method that takes in difficulty level $d$ and randomly generates $n$ instances that define problems of that difficulty.

These components work together by defining a self-contained problem generator: $\rho_0$ produces structured instance parameters, $O$ turns these parameters into a natural language prompt for the solver, and $R$ evaluates any proposed solution against the underlying parameters. This design ensures that all instances of a task share the same verification logic, while allowing unlimited procedural variation in the questions themselves.

**3. LLM-as-a-Judge.** We employ two stages of LLM evaluation to filter generated environments. The first stage evaluates the code implementation of $R$ and $O$ with emphasis on two key criteria. The first, *reference-free verification*, requires $R$ to verify solutions without access to a reference answer, exploiting the generator-verifier gap when possible. The second criterion, *computational advantage*,

**ReSyn Environment Example: Grid Path Cost Optimization**

**Instance Generation ($\rho_0$):**

```python
def generate_instance(difficulty):
  s = difficulty + 2
  grid = [[randint(1, 9) for _ in range(s)]
        for _ in range(s)]

  return {
    'grid': grid, 'size': s,
  })
```

**Sample Instance (difficulty=2):**

```
grid: [[2, 7, 3, 1], [5, 1, 8, 4], [3, 6, 2, 9], [1, 4, 7, 2]]
size: 4
```

**Generated Question:**

```
Find minimum cost path from top-left to bottom-right
(only right/down moves):
2 7 3 1
5 1 8 4
3 6 2 9
1 4 7 2
Answer: <answer>NUMBER</answer>
```

**Reward Function ($R$):**

```python
def solve_grid(grid: List[List[int]]):
  # Helper to solve grid via dynamic programming
  n = len(grid)
  dp = [[float('inf')] * n for _ in range(n)]
  dp[0][0] = grid[0][0]

  for i in range(0,n):
    for j in range(0,n):
      if i > 0:
        dp[i][j]=min(dp[i][j],
                     dp[i-1][j]+grid[i][j])
      if j > 0:
        dp[i][j]=min(dp[i][j],
                     dp[i][j-1]+grid[i][j])
  return dp[n-1][n-1]

def verify(response, instance):
  match = re.search(r'<answer>(\d+)</answer>',
      response)
  if not match:
    return False

  answer = int(match.group(1))
  return answer == solve_grid(instance['grid'])
```

Note: Code has been restructured for space with no change to functionality.

Figure 2: Example synthetic environment generated by the ReSyn pipeline.

seeks tasks that can be efficiently solved with computational tools (e.g., graph algorithms, constraint solvers) but are challenging to solve by hand. Environments must pass at least one of these key criteria, along with basic requirements for implementation completeness and proper difficulty scaling.

The second stage examines NL questions at each difficulty level, denoted $Q = O(s)$; $s \sim \rho_0(d)$. It assesses whether they are well-specified (providing complete context, clearly defined operations, and unambiguous goals) and free from logical loopholes that could bypass the intended reasoning process. For failed environments, we maintain a record of issues identified by the LLM judge and provide them with the original implementation to be revised. Revised environments are evaluated by the same process again and discarded if they still fail.

**4. Instance Generation.** For each surviving environment, we sample a fixed number of problem instances $\{s\}_{i=1}^n = \rho_0(d, n)$ for each difficulty level $d \in \{1, \ldots, 5\}$. We create training-ready $(Q, V)$ pairs from each $s$ by using the reward and observation functions:

$$(Q, V) = \big(O(s), R(s, \cdot)\big)$$

Illustrated in Figs 2, this step is entirely procedural and does not require additional LLM queries.

**5. LLM Solving & Difficulty Calibration.** Finally, we prompt the LLM to generate solutions for each question $Q$: $a \sim p_{\text{LLM}}(\cdot \mid Q)$, and compute scores by calling the verifier on $V(a)$. From these results, we compute the solve rate at each difficulty level, producing an accuracy-difficulty curve per environment. To retain only environments whose difficulty parameter $d$ meaningfully controls problem hardness, we test for a significant negative correlation between solve rate and difficulty level using a one-sided Wald test with $\alpha = 0.05$. Environments failing this test – often because they are trivial (near-100% accuracy across all levels) or impossible (0% accuracy) – are removed.

The final output is a collection of parameterized reasoning environments $\{E_i\}$ like the one shown below, and a pool of procedurally generated question-verifier pairs $\{(Q, V)_j\}$. This design supports RLVR by enabling large-scale data generation and fine-grained control over both diversity and difficulty. All stages of the pipeline use Claude 3.5 Sonnet v2 as the LLM.

## 3 EXPERIMENTS

We construct the ReSyn dataset using the pipeline detailed in §2. Starting with around 100 keywords, we generate 8 environments per keyword, out of which 418 distinct environments survive all filtering stages. We run the procedural generation logic in each environment to generate a total of 16K training instances and 500 validation instances. As described above, each instance is a

$(Q, V)$ pair where $Q$ is a natural language question and $V$ is a verifier implemented in code. While the pipeline can in principle produce far more environments, we match the dataset scale of prior work (Liu et al., 2025a) and remain within available training compute. We perform an analysis in Appendix A.5, indicating substantially higher data diversity in ReSyn than SynLogic.

## 3.1 REINFORCEMENT LEARNING WITH VERIFIABLE REWARDS

Reinforcement learning with verifiable rewards (RLVR) provides a mechanism for training a policy model $\pi_\theta$ using the dataset of $(Q, V)$ pairs. At each iteration, a question $Q$ is sampled and presented to the model. The model then generates candidate solutions $a_1, \ldots, a_G \sim \pi_\theta(\cdot \mid Q)$.

The verifier $V$ associated with $Q$ evaluates each solution and returns a binary reward signal

$$r_i = V(a_i)$$

where $r_i = 1$ if the solution is judged correct and $0$ otherwise. A format reward is sometimes also added to encourage certain response structures.

The collected rewards are then used to update the model parameters $\theta$ by taking gradient steps against a reinforcement learning objective $\mathcal{L}(\pi_\theta, Q, \{a_i\}_{i=1}^G, \{r_i\}_{i=1}^G)$. In general, this objective encourages $\pi_\theta$ to increase the likelihood of solutions that receive positive reward and decrease the likelihood of those that are incorrect. Over repeated interactions, this process, illustrated in Fig. 3, aligns the model toward solutions that pass the verifiers across the synthetic environments.

## 3.2 TRAINING DETAILS

Our policy model is initialized from `Qwen2.5-7B-Instruct` (Team, 2024). We do not initialize from the base model `Qwen2.5-7B` to avoid having to re-learn basic instruction following and output formatting. This also helps ensure that evaluation metrics can reflect genuine gains in reasoning performance rather than format adherence.

We train this model on RESYN data using the open-source DAPO recipe (Yu et al., 2025). Although our method could in theory work with many similar RL algorithms, we use DAPO because it has been shown to be effective in similar settings. Unless otherwise noted, we train all models for 400 update steps with default hyperparameters.

Each prompt $Q$ is prefixed with explicit instructions to place intermediate reasoning inside `<think>...</think>` tags and the final answer inside `<answer>...</answer>` tags, following conventions from prior work (DeepSeek-AI, 2025). The exact prompt prefix is provided in A.4.

Following these conventions, the reward is computed as the product of two components:

- **Format score:** a binary indicator of whether the output follows the required tag structure.
- **Answer score:** equal to $V(\text{extract}(a_i))$.[1] We first extract from `<answer>` tags and then apply the question-specific verifier.

## 4 MAIN RESULTS

We evaluate `Qwen2.5-7B-ReSyn` and `Qwen2.5-7B-Instruct` on a suite of reasoning and math benchmarks, including Big-Bench Hard (BBH; Suzgun et al. (2022)), Big-Bench Extra Hard (BBEH; Kazemi et al. (2025)), GSM8K (Cobbe et al., 2021), and AIME 2024 (Finkelstein et al., 2024). For all benchmarks, we sample with temperature $0.8$ and top-$p = 0.95$, which helps mitigate occasional output degeneration observed in both models.

The overall results are reported in table 1. Across all evaluated benchmarks, ReSyn consistently outperforms the Instruct baseline, including math benchmarks, where no real domain-specific data was provided. These results support the effectiveness of learning general reasoning skills from synthetic verifier-based data.

---

[1]Since $V$ is implemented by an LLM, it may occasionally throw errors; in such cases, we assign a return value of 0. After our filtering stages, the error rate is negligible. Moreover, the dynamic sampling mechanism (Yu et al., 2025) in DAPO ensures that verifiers that consistently fail are ignored in model updates.

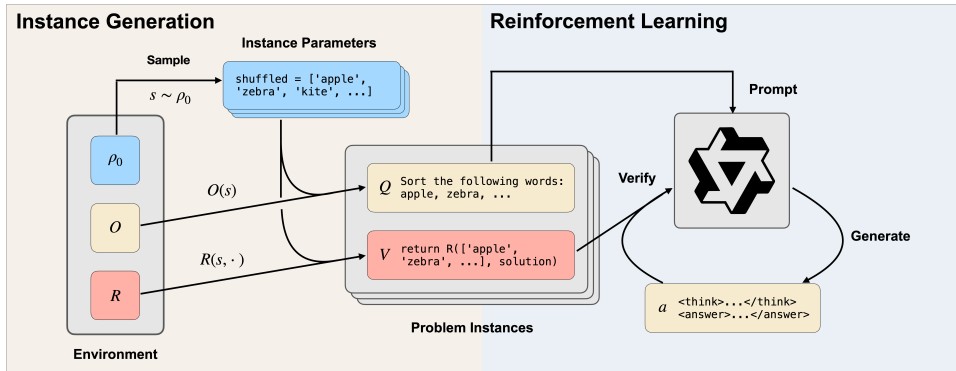

Figure 3: Overview of the ReSyn training pipeline. **Left (Instance Generation)**: Each environment generates instances from $\rho_0$, which are transformed by the observation function $O$ and reward function $R$ into questions and verifiers $(Q, V)$. **Right (Reinforcement Learning)**: A policy model generates candidate solutions for $Q$, which are evaluated by $V$ to provide rewards for model updates.

| Model | #Params | BBH (zero-shot) mean@4 | BBEH mean@4 | GSM8K-test mean@4 | AIME 2024 mean@128 |
|---|---|---|---|---|---|
| **Qwen-2.5-Instruct** | 7B | 65.9 | 11.2 | 82.3 | 9.8 |
| **SynLogic*** | 7B | 66.5 | 8.0 | — | 10.0 |
| **ReSyn** | 7B | **75.2** | **14.3** | **91.4** | **14.0** |
| **Llama-3.1-Instruct** | 8B | 44.7 | 8.2 | 70.7 | 3.5 |
| **Mistral-Instruct** | 7B | 28.2 | 6.4 | 22.3 | 0.1 |

Table 1: Evaluation of ReSyn versus the base Instruct model across reasoning and math benchmarks. All benchmarks are evaluated in zero-shot conditions using temperature $0.8$ and top-$p$ $0.95$ sampling. *Performance of SynLogic-7B is taken as reported in Liu et al. (2025a).

**Baseline Selection**: TinyZero (Pan et al., 2025) and Logic-RL (Xie et al., 2025) also apply synthetic data generation for RL, but both methods train on a single task only. Notably, TinyZero's sole training task (Countdown) is included in the SynLogic suite. Thus, we compare to SynLogic (Liu et al., 2025a) as the strictly stronger baseline among these works.

## 4.1 BIG-BENCH HARD (BBH)

The Big-Bench Hard benchmark consists of 23 tasks spanning topics in logical and commonsense reasoning, with many requiring multi-step reasoning and task-specific strategies.

In initial experiments, we found that the performance of `Qwen2.5-7B-Instruct` on BBH is highly dependent on imitating few-shot CoT examples. To disentangle reasoning skill – i.e., the ability to devise reasoning steps for an unseen task – from in-context learning ability, we evaluate both models under 0-, 1-, and 3-shot settings. We also find that Liu et al. (2025a) under-reports the performance of `Qwen2.5-7B-Instruct` on BBH due to answer extraction issues, so we fix this and provide details in Appendix A.2.

The results are presented in Table 2. Across all few-shot settings, ReSyn outperforms Instruct. Per-task accuracies show that gains are most significant on logical reasoning tasks, such as `logical_deduction_*_objects`, `temporal_sequences`, and `web_of_lies`. Notably, 0-shot ReSyn exceeds 3-shot Instruct by nearly 5%, suggesting that RL training on ReSyn yields reasoning ability that can surpass in-context learning from demonstrations. However, unlike Instruct, ReSyn's performance does not increase monotonically with more in-context examples. We hypothesize that imitating provided CoT demonstrations can be less effective than reasoning in the model's own preferred style for some tasks.

| Model | 0-shot | 1-shot | 3-shot |
|---|---|---|---|
| Instruct | $65.9 \pm 0.5$ | $67.1 \pm 0.5$ | $70.4 \pm 0.5$ |
| ReSyn | $75.2 \pm 0.5$ | $71.9 \pm 0.5$ | $73.2 \pm 0.5$ |

Table 2: BigBench-Hard overall accuracy (%) under different prompting setups.

## 4.2 BigBench Extra Hard (BBEH)

BBEH is a more challenging benchmark for evaluating general reasoning in large language models. It replaces each task in BBH with a newly designed task that tests the same underlying reasoning skill but at a substantially higher difficulty level. Notably, many smaller models perform near chance level on this benchmark, showing large room for improvement.

Given the low accuracy of models around the 7B scale, we also evaluate two trivial baselines:

- **Chance**: randomly selects the ground-truth answer from *any* problem within the same task.
- **Majority**: always outputs the most common ground-truth answer for the task.

Table 3 reports overall accuracies. Instruct underperforms the Majority baseline, highlighting the difficulty of BBEH. ReSyn achieves $14.29\%$ accuracy – although the absolute number is low, this gain represents a meaningful relative improvement of $\sim 27\%$ over the Instruct model, suggesting that training on our synthetic data generalizes to harder reasoning problems in BBEH.

| Model | #Params | mean@4 |
|---|---|---|
| Chance | — | $8.83 \pm 0.2$ |
| Majority | — | $13.1 \pm 0.5$ |
| Qwen 2.5-Instruct | 7B | $11.2 \pm 0.4$ |
| ReSyn | 7B | **$14.3 \pm 0.4$** |

Table 3: BigBench Extra Hard overall accuracies (%, micro-average).

| | Chance | Majority | Instruct | ReSyn |
|---|---|---|---|---|
| Chance | – | 0 | 3 | 2 |
| Majority | 8 | – | 9 | 6 |
| Instruct | 9 | 3 | – | 1 |
| ReSyn | 11 | 7 | 6 | – |

Table 4: Number of BBEH tasks (out of 23) where the row model significantly outperforms the column model ($\alpha = 0.01$, paired bootstrap test).

It is possible that large improvements are realized on a subset of tasks, while other tasks remain out of reach. To examine this, we compare models on a per-task basis using paired bootstrap tests ($\alpha = 0.01$) for each pair of models. Table 4 reports, for each row–column pair, the number of tasks (out of 23) on which the row model significantly outperforms the column model. ReSyn outperforms Instruct on 6 tasks (vs. 1 in the opposite direction) and increases the number of tasks above Chance ($9 \rightarrow 11$) and Majority ($3 \rightarrow 7$), showing consistent gains on tasks in the benchmark. Names of the exact tasks and per-task accuracy comparisons are provided in Appendix A.3.

## 5 Ablation Studies

### 5.1 Ablation: The Generator-Verifier Gap

A central hypothesis of our work is that synthesizing verifier-based data can provide more effective supervision than generating solution data. This is an example of the *generator–verifier gap*: when generating synthetic solution data, the LLM must solve each problem correctly at generation time, whereas with verifier-based data, the LLM need only specify the rules for checking a solution. This allows the generator to specify challenging tasks without being constrained by its own problem-solving capabilities. To test this hypothesis, we compare our method against ablated versions that do not use verifiers or do not use code:

- **Verifier-RL (Ours)**: The proposed method, where the LLM generates verifiers, which are used to compute rewards during RL.
- **Code-RL (Ours)**: An alternative method, where the LLM generates and executes code to obtain a reference answer. We then use answer matching rewards during RL.

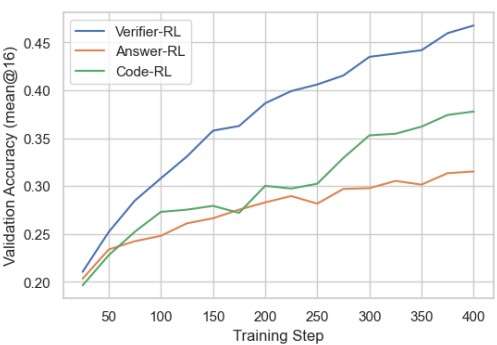

Figure 4: Validation accuracy vs. training step for the three RL-based ablations, computed on `ReSyn-val` and averaged over 16 samples per problem.

| Method | BBH | BBEH |
|---|---|---|
| Verifier-RL | **75.24** | **14.61** |
| Code-RL | 74.94 | 14.24 |
| Answer-RL | 68.83 | 14.33 |

Table 5: Final BBH and BBEH accuracies (%) for all ablation settings.

| Configuration | BBH mean@4 |
|---|---|
| ($N$=400, $M$=40) | 75.19 |
| ($N$=100, $M$=160) | 69.85 |
| ($N$=25, $M$=640) | 71.20 |

Table 6: Dataset scaling ablation: number of environments ($N$) vs. instances per environment ($M$) with dataset size fixed at $N \times M \approx 16K$.

- **Answer-RL**: The method most similar to prior work, where the LLM generates reasoning and answers for each problem. We then use answer matching rewards during RL.

In all cases, LLM-generated verifiers or solutions may produce incorrect rewards. We allow such errors to occur in our comparison, since a part of our claim is that generating verifiers should be less error-prone than generating solutions.

We compare learning dynamics for the three settings in Figure 4. We observe that validation accuracy rises fastest for Verifier-RL, followed by Code-RL, and then Answer-RL, suggesting that verifier and code-based rewards provide a stronger and more reliable learning signal during RL training. This also results in a significant difference in downstream performance on BBH: with the base model at 65.9, both code-based methods achieve ≈14% relative improvement compared to about 4% by Answer-RL. Overall, **Verifier-RL** seems to provide the highest quality supervision, achieving the best performance as measured by the in-domain validation set and reasoning benchmarks.

## 5.2 ABLATION: SCALING ALONG TASKS VS. INSTANCES

An advantage of our synthetic data generation method, compared to prior approaches, is that it can produce entirely new task structures rather than only procedurally generating instances of fixed tasks. This enables scaling the dataset along two axes:

1. **Task diversity**: leveraging the breadth of LLM knowledge to synthesize new reasoning environments with different structures and reasoning patterns.

2. **Instance count**: procedurally generating more instances per environment, which can still provide new learning signal by requiring reasoning over more information or longer reasoning chains.

To study how scaling along these two axes affects performance, we construct training sets containing varying numbers of environments $N$ and instances per environment $M$ while controlling for the total dataset size $N * M$. We keep $N * M$ around 16K and train models with $(N, M) \in \{(400, 40), (100, 160), (25, 640)\}$. For each $(N, M)$ configuration, we randomly sample $N$ environments from the full ReSyn dataset and generate $M$ instances per environment, uniformly split across difficulty levels 1-5. For the $N = 400$ dataset, we also perform an analysis in Appendix A.5 comparing its diversity to that of SynLogic.

For each configuration, we train a model using the same setup as in our main experiments and evaluate it on BBH. The results for all configurations are shown in Table 6: being able to generate a large collection of tasks is crucial to ReSyn's performance on the benchmark. Together, these ablations indicate that verifier-based supervision and task diversity are the key drivers of ReSyn's effectiveness in training stronger reasoning models.

## 6 RELATED WORKS

**Training Large Language Models for Reasoning:** In recent years, there has been significant interest in training large language models (LLMs) to perform reasoning-intensive tasks, particularly through reinforcement learning (RL). DeepSeek-AI (2025) demonstrated that large-scale RL, even when guided only by correctness and format rewards, can yield state-of-the-art performance on math and coding benchmarks. This setup also produced long chains of thought (CoTs) and behaviors resembling self-verification and reflection. Subsequent research has worked on improving the training dynamics of this recipe (Yu et al., 2025; Liu et al., 2025b).

**Synthetic Data Generation for Reasoning:** Early work has demonstrated that training on model-generated data can effectively improve instruction-following ability (Wang et al., 2023; Taori et al., 2023). Recent research has increasingly turned toward using synthetic data to teach models reasoning skills. Approaches in this space can be broadly categorized into several categories:

- **Procedural Generation**: Several works train models with RL on procedurally generated problems from manually designed task templates (Pan et al., 2025; Xie et al., 2025; Liu et al., 2025a), showing that training on a small number of tasks can already elicit emergent reasoning behaviors and transfer to real benchmarks such as competition math. Works have created datasets of up to 100 handcrafted logical reasoning tasks to support training and evaluation (Stojanovski et al., 2025).
- **Model-based Generation**: These methods use LLMs to synthesize problems and solutions, allowing greater diversity than procedural templates. Havrilla et al. (2025) iteratively calls an LLM to mutate math problems, following a quality-diversity algorithm, while Guo et al. (2025) prompts with retrieved passages to generate QA data for diverse domains. Project Loong (CAMEL-AI.org, 2025) introduces code-based solutions to verify correctness, improving the reliability of synthetic labels.
- **Environment Generation**: Instead of producing problem–solution pairs, some recent work defines environments with rule-based success criteria. Verma et al. (2025) introduces a collection of board, number, and card games to benchmark LLM reasoning by pitting them against RL-trained agents. Zhou et al. (2025) has language model agents propose challenges for themselves (e.g., web browsing tasks), where success criteria are specified via code-based verifiers.

The works most related to ours are SynLogic (Liu et al., 2025a) and Synthetic Data RL (Guo et al., 2025). **SynLogic** relies on 35 manually curated tasks; we instead scale task creation by over an order of magnitude and achieve much stronger results on BBH and BBEH. **Synthetic Data RL** generates domain-specific QA pairs; we instead match its GSM8K performance (91.4% vs. 92.1%) by generating code-based reasoning environments to provide more reliable rewards for RL.

## 7 CONCLUSION

We introduced RESYN, a pipeline for synthesizing large collections of reasoning environments equipped with code-based instance generation and verifiers, enabling reinforcement learning with verifiable rewards (RLVR) at scale. Unlike prior work that relies on small sets of manually curated tasks or model-generated solutions, our approach leverages the generator–verifier gap to construct diverse and challenging tasks that remain straightforward to verify.

Training models on the ReSyn dataset with open RL recipes yields consistent improvements across reasoning and math benchmarks, including a $+14\%$ relative improvement on BBH and a $+27\%$ relative improvement on BBEH. Ablation studies demonstrate that verifier-based supervision provides more reliable reward signals than solution-based supervision, and that scaling the number of environments is a particularly effective way to elicit general reasoning skills. These results highlight verifier-driven synthetic data as a scalable foundation for reasoning-focused LLM training.

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

## A APPENDIX

### A.1 RESYN KEYWORDS

*Array traversal, Backtracking, Boolean Evaluation, Boolean Logic, Chain of Dependencies, Circuit Design, Connected Components (graph), Constraint Satisfaction, Coordinate System, Counting, Custom Operators, Date Calculation, Deductive Reasoning, Direction Tracking, Dyck Words, Enumeration, Expression Evaluation, Expression Transformation, First-Order Logic, Geometry, Grid, Grid Search, Grid Traversal, Information Extraction, Information Retrieval, Interval Analysis, Knowledge Base, Lexicographical Order, Linear Ordering, Linear Search, Logic Expressions, Math Operations, Matrix, Modal Logic, Number Theory, Order of Operations, Parentheses Matching, Path Finding, Pattern Matching, Pattern Recognition, Permutation, Permutation Cipher, Position Tracking, Propositional Logic, Rotation, Rule-based Reasoning, Sequence Arrangement, Set Classification, Set Theory, Sorting, Stack, State Transition, String Manipulation, String Matching, Table Analysis, Time Scheduling, Topological Sort, Transposition Cipher, Truth Table, Word Search, Shortest Paths, Connected Components, Stable Matching, Dynamic Programming, Recursion, Greedy Algorithms, Divide and Conquer, Breadth-First Search, Depth-First Search, Path Optimization, Minimum Spanning Tree, Network Flow, Topological Sorting, Sliding Window, Union Find, Priority Queues, Linear Programming, Tree Traversal, Graph Coloring, Knapsack Problem, Combinatorial Optimization, Cycle Detection, Interval Scheduling, Minimum/Maximum Flow, Edit Distance, Euler Tours, Traveling Salesman, Longest Common Subsequence, Longest Increasing Subsequence, Item Assignment, Boolean Satisfiability, Tabular Data.*

### A.2 BBH EVALUATION DETAILS

We find that the BBH accuracies reported for Qwen2.5-7B and Qwen2.5-7B-Instruct in Liu et al. (2025a) are underestimated due to issues with answer extraction. The default answer extraction logic searches for fixed phrases such as "So the answer is X," which often fails due to minor variations in phrasing. To address this, we (1) modify the prompt to instruct the model to enclose its answer in <answer>...</answer> tags and provide an example answer from the task, and (2) relax the matching logic to accept direct statement of the answer (e.g., "Alice" instead of "(A)" or "(A) Alice"). With these changes, three-shot accuracy for Qwen2.5-7B-Instruct rises from 62.7% (as reported by Liu et al. (2025a)) to 70.4%.

### A.3 BBEH SUBTASK PERFORMANCE

**Pairwise Per-Task Comparisons:** The grid below shows the tasks from BBEH where the row model outperforms the column model, tested for statistical significance using a paired bootstrap test ($\alpha = 0.01$).

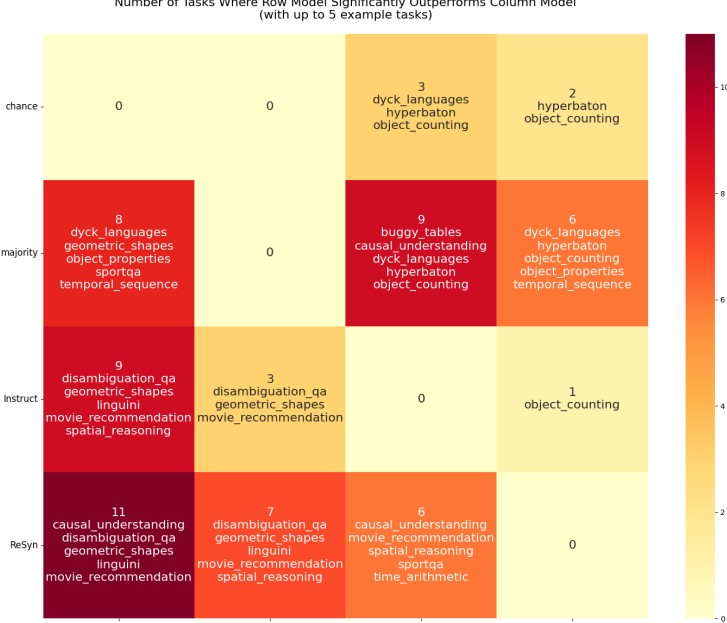

**Per-Task Model Performance**: Comparison of `Qwen2.5-7B-Instruct` and `ReSyn-7B` on BBEH. Among the 23 tasks, we display only those where at least one model reaches $\geq 5\%$ accuracy, since many tasks are out of reach for models of this scale. `ReSyn-7B` delivers consistent gains and achieves substantial improvements on several subtasks (e.g., $+10\%$ on `causal_understanding`).

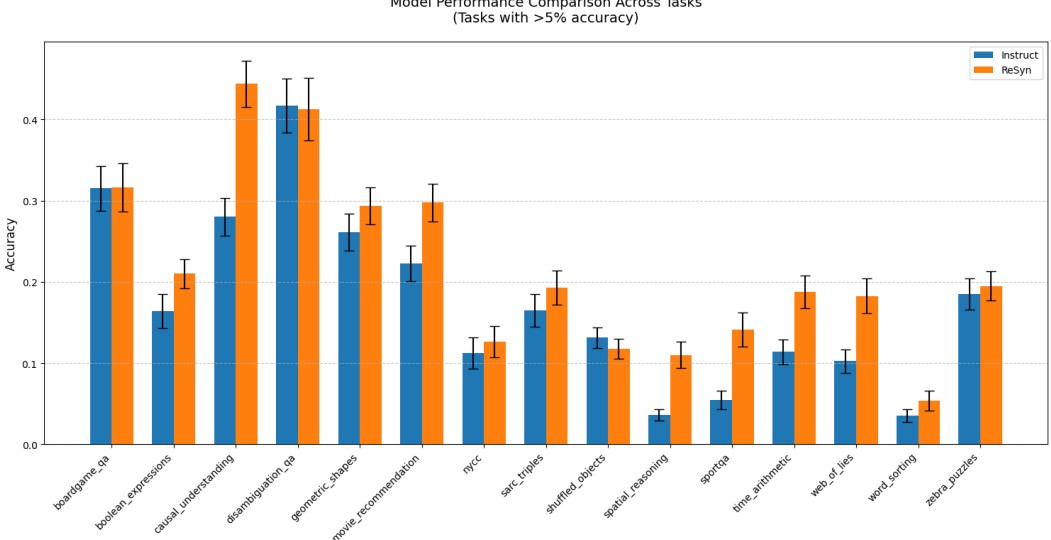

### A.4 PROMPT PREFIX

During training and inference, we prepend this string to prompts to encourage the model to generate responses with a specific format.

```
1 Solve the following problem step by step. First, think about the
     reasoning process in the mind and then provide the answer. The
     reasoning process is enclosed within <think> </think> and the final
     answer is enclosed within <answer> </answer> tags, respectively, i.e
     ., <think> reasoning process here </think> <answer> answer here</
     answer>.
```

## A.5 DATASET DIVERSITY ANALYSIS

The ReSyn pipeline builds a large and diverse set of verifiable tasks starting from only a few seed dozen keywords. To explore the difference in diversity of core reasoning tasks between ReSyn and comparable dataset methods like SynLogic, we introduce a mechanism for extracting and measuring the entropy of tasks in a dataset.

We draw inspiration from semantic entropy (Farquhar et al., 2024) and develop an *LLM-assisted task entropy* pipeline that runs the following steps: (1) Sample 1000 tasks randomly from each dataset to ensure fair comparison, (2) Generate detailed semantic descriptors for each task using a strong LLM with prompts that capture specific reasoning patterns and constraints, (3) Embed descriptors using sentence-transformers (all-mpnet-base-v2) to create semantic representations, (4) Apply hierarchical agglomerative clustering with cosine distance and configurable similarity thresholds, (5) Calculate Shannon entropy across cluster distributions to quantify semantic diversity.

**Formal Algorithm Definition.** Let $D = \{d_1, d_2, \ldots, d_n\}$ be a set of task descriptors generated by the LLM. We compute semantic embeddings $E = \{e_1, e_2, \ldots, e_n\}$ where $e_i = \text{SentenceTransformer}(d_i) \in \mathbb{R}^{768}$. For clustering, we define the cosine distance between embeddings as:

$$\text{dist}(e_i, e_j) = 1 - \frac{e_i \cdot e_j}{\|e_i\|\|e_j\|} \tag{1}$$

We apply hierarchical agglomerative clustering with single linkage and distance threshold $\tau$, producing clusters $C = \{C_1, C_2, \ldots, C_k\}$ where tasks are grouped if their minimum pairwise distance is below $\tau$. The semantic entropy is then calculated as:

$$H = -\sum_{i=1}^{k} p_i \log_2(p_i) \tag{2}$$

where $p_i = |C_i|/n$ is the proportion of tasks in cluster $C_i$. Higher entropy thus indicates greater task diversity. We plot the semantic entropy as a function of $\tau$ for ReSyn and SynLogic below. We measure consistently higher semantic entropy in ReSyn.

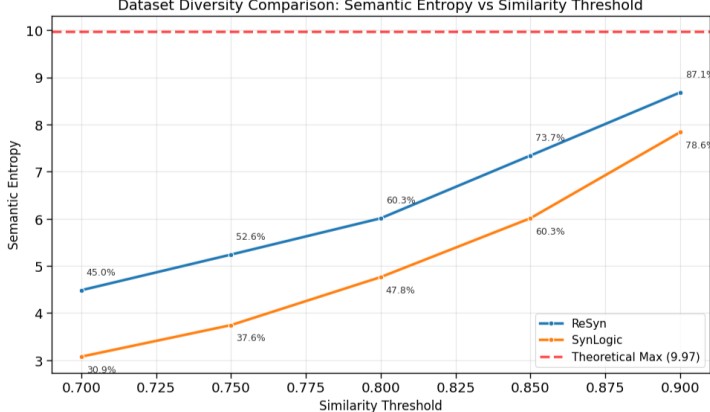

For example, our LLM generates descriptors such as *"Group assignment optimization with cardinality bounds, pairwise conflict constraints, and complete partition requirements"* for constraint satisfaction tasks, *"Graph vertex coloring with adjacency-based color exclusion, chromatic number constraints, and complete vertex coverage requirements"* for graph theory problems, and *"Modal*

*logic formula evaluation in Kripke structures with accessibility relations, necessity/possibility operators, and world-based truth valuation"* for logical reasoning tasks. We find through qualitative analysis that descriptors with a sentence transformer embedding cosine similarity around 0.8 or higher tend to be the same core task. At this threshold, tasks with genuinely similar reasoning patterns cluster together appropriately. For example, one cluster contains tasks described as *"Activity selection with temporal non-overlap constraints using greedy earliest-finish-time optimization for maximal scheduling"*, *"Weighted interval scheduling with maximal non-overlapping subset selection and greedy earliest-deadline optimization"*, and *"Interval scheduling maximization - selecting maximum non-overlapping time intervals using greedy algorithm with earliest finish time priority"*, representing legitimate variations of the same core scheduling optimization problem.

### A.5.1 CORE TASK DESCRIPTORS PROMPT

To extract task descriptors, we used the following prompt to Claude 4 Sonnet:

```
1  You are an expert at analyzing reasoning tasks and categorizing them by
        their specific logical patterns and problem structures.
2
3  {task_section}
4
5  TASK: For each task, provide a detailed descriptor that captures the
        specific core reasoning pattern and problem structure.
6
7  INSTRUCTIONS:
8  1. Describe the exact logical operations required (e.g., "Grid
        pathfinding with sum constraints", "Sequence alternation pattern
        detection", "Combinatorial placement with mutual exclusion")
9  2. Include key constraints and problem mechanics, not just high-level
        categories
10 3. Distinguish between tasks that might share keywords but have different
         reasoning patterns
11 4. Be specific enough to differentiate similar-seeming tasks
12 5. Focus on what makes each task's reasoning unique
13
14 RESPONSE FORMAT:
15 Return a JSON object where each key is the task identifier and the value
        is the detailed descriptor.
16
17 Example:
18 {{
19   "TASK_0": "Grid pathfinding with cumulative sum optimization and
        directional movement constraints",
20   "TASK_1": "Sequential pattern recognition with alternating increase-
        decrease validation",
21   "TASK_2": "Constraint satisfaction with backtracking and mutual
        exclusion rules"
22 }}
23
24 RESPONSE:
25
```

### A.6 LLM USAGE

In the writing of this paper, a large language model was used as a writing assistant. It was mainly used for functions such as rephrasing or shortening text, drafting section introductions and figure captions, suggesting formatting in LaTeX, and generally polishing the style and flow of writing.

