# OpenReview forum: "ReSyn: Autonomously Scaling Synthetic Environments for Reasoning Models"
_ICLR.cc/2026/Conference — Submitted to ICLR 2026_

### Official Review · Reviewer_o8zE · 2025-10-30

**Soundness:** 2
**Presentation:** 2
**Contribution:** 2
**Rating:** 2
**Confidence:** 4

**Summary:**

This paper introduces RESYN, a scalable pipeline that autonomously generates diverse reasoning environments, where each environment includes a problem generator and a code-based verifier to enable training via reinforcement learning. By leveraging the generator-verifier gap and scaling task diversity, a model trained on RESYN data achieves significant improvements on reasoning benchmarks, including a 27% relative gain on the challenging BBEH benchmark.

**Strengths:**

1. The target of this paper for automatically constructing tasks for training LLMs is critical.
2. The proposed method achieves improved performance compared to baselines.

**Weaknesses:**

1. The paper claims the proposed method achieves OOD improvement on Big-Bench Hard (BBH). However, in section 2.2, the paper uses subtasks of BBH as input for constructing the training dataset. This may lead to data leakage, weakening the claim of OOD evaluation.
2. The diversity of environments is not validated. The pipeline of this framework fully relies on Claude 3.5 to evaluate and filter the generated environments, and generate from 100 keywords. This process may incur mode collapse, which means the 418 tasks share similar logic or patterns. Therefore, the diversity of tasks is needed to validate and analyze.
3. The proposed method only reports 418 survived environments, but does not provide the pipeline or method for how many other environments are filtered out and why they are filtered.
4. The claim of generalization to math benchmarks (like GSM8K) is invalid. The paper claims this is an out-of-domain gain. However, the seed keyword list in Appendix A.1, used to generate the environments, is heavily populated with explicit mathematical and algorithmic concepts such as "Number Theory," "Math Operations," "Dynamic Programming," and "Combinatorial Optimization". Therefore, the strong performance on GSM8K is an expected "in-domain" transfer from synthetic algorithmic tasks, not a proof of general reasoning transfer.

**Questions:**

See weaknesses.

---

> ### Author Response · Authors · 2025-11-24
>
> We thank the reviewer for the detailed feedback.  We will address the weaknesses and questions in order:
>
> **Data leakage**
> The specific choice of BBH as a seed source is not essential to our approach. Alternative methods (e.g., having an LLM propose keywords directly or manually constructing a keyword set) would also likely be viable. Moreover, the chance of data leakage is minimal, since we only use keywords and no other information about the original data as input to our generation pipeline. The keywords are listed in the appendix — they are simple phrases and do not contain any details from the benchmarks.
>
> **Task diversity**
> To compare the diversity of our dataset with SynLogic, we conduct an analysis based on semantic clustering. Our analysis is included in the new appendix A.5.
>
> **Data filtering**
> Overall, around 50% of the generated environments are kept. We will include the exact number kept after each stage in our revisions.
>
> **Claim about OOD transfer**
> We agree that GSM8K might be partly in-domain, as its problems are relatively simple and can resemble patterns produced by our synthetic environments. However, we also observe improvements on AIME (+40% relative improvement), which consists of competition-style math problems requiring specialized, complex reasoning. These problems differ substantially in structure and difficulty from the synthetic tasks generated by our pipeline, making accidental overlap unlikely and suggesting genuine transfer to mathematical reasoning.

---

### Official Review · Reviewer_xoLM · 2025-11-01

**Soundness:** 3
**Presentation:** 3
**Contribution:** 2
**Rating:** 4
**Confidence:** 5

**Summary:**

This paper proposes generating diverse environments using large language models (LLMs) instead of reasoning traces, along with verifiers implemented in code to create question–verifier (Q, V) pairs. The core idea is that verifier-based supervision offers stronger learning signals than solution-based data. Unlike previous work that relies on hand-picked problems, this method automatically generates Q, V pairs via LLMs.

**Strengths:**

1. The approach demonstrates impressive improvements (Table 4) over both the instruct model and baselines across four datasets.
2. The dataset generation strategy integrates multiple filtering mechanisms and appears methodologically sound.
3. The framework allows dynamically generated tasks rather than fixed ones as in prior work, which contributes to performance gains.

**Weaknesses:**

1. There is substantial variance when the number of environments or tasks changes. It remains unclear whether the upward scaling trend holds beyond 400 environments, as performance sometimes falls below prior work (e.g., on BBH).
2. The set of baselines is limited. Only **SynLogic** (Liu et al.) is included. The authors should also compare against **TinyZero** (Pan et al.), **Logic-RL** (Xi et al.), and **Synthetic Data RL** (Guo et al.) to contextualize performance more broadly. Another possible candidate: https://arxiv.org/abs/2505.24760 (NeurIPS 2025)
3. The advantages of the proposed method over **Synthetic Data RL** (Guo et al.) are not mentioned. Since Synthetic Data RL can also scale with many task definitions, a direct comparison would strengthen the paper’s claims.
4. The effect of the RL training algorithm beyond **DAPO** is unexplored. As prior work (e.g., Liu et al.) uses **GRPO**, it is unclear whether improvements stem from the algorithm or from the proposed synthetic data.

**Questions:**

See Weaknesses above.

---

> ### Author Response · Authors · 2025-11-24
>
> We thank the reviewer for the detailed feedback.  We will address the weaknesses and questions in order:
>
> **Limited baselines:**
> Thank you for pointing out these relevant related approaches. We are aware of TinyZero and Logic-RL as relevant prior work applying RL to synthetic reasoning tasks. However, both methods train on a single task, whereas SynLogic employs a handcrafted suite of roughly 30 tasks. Notably, TinyZero’s sole training task (Countdown) is already included in the SynLogic benchmark. Therefore, SynLogic serves as a strictly stronger and more general baseline than TinyZero and Logic-RL. Since our method already outperforms SynLogic, we did not include separate comparisons to these single-task baselines. We have included this discussion in the updated draft.
>
> **Comparison to Synthetic Data RL:**
> Indeed, Synthetic Data RL is one of our closest comparisons other than SynLogic. An essential advantage of our method over theirs is that we generate code to act as verifiers, whereas they generate answers directly in natural language. If we controlled for the data domain, Synthetic Data RL would be most similar to Answer-RL in our ablation experiment in 5.1, which indicates that our code-based method yields better downstream performance.
>
> **DAPO vs GRPO**
> Our most direct comparison is the models trained by SynLogic (Liu et al.) which also uses DAPO;, so the comparison is fair.

---

### Official Review · Reviewer_7VmH · 2025-11-01

**Soundness:** 1
**Presentation:** 1
**Contribution:** 2
**Rating:** 2
**Confidence:** 4

**Summary:**

This paper proposes ReSyn, a pipeline that automatically generates synthetic reasoning environments with code-based verifiers and instance generators. The authors argue this approach can scale “reasoning-focused” reinforcement learning with verifiable rewards (RLVR) beyond a small number of handcrafted tasks.

**Strengths:**

- The paper is clearly motivated by the recent trends in RLVR.

- The idea of combining automatic environment synthesis with code-based verifiers is conceptually appealing.

- The authors include ablations on verifier vs. answer-based supervision and task vs. instance scaling.

**Weaknesses:**

1. **Lack of experimental rigor and details.** The experimental section omits key information—how exactly were the questions and environments generated from keywords, what proportion were filtered out, and how many survived each stage? The authors mention using “Claude 3.5 Sonnet v2” but do not provide prompts, seed examples, or reproducibility details. It is also unclear whether any existing datasets (e.g., BBH templates) were reused or rephrased.


2. **Minimal improvement from baselines.** The reported gains are modest. For instance, the BBEH improvement from 11.2 → 14.3 is still near chance level. There is no comparison with strong recent baselines such as R1-Zero-like methods, or self-play verifiers.

3. **Unclear generalizability.** The environments appear narrow—mostly code-style puzzles or rule-based tasks (Appendix A.1). It remains unclear whether the learned skills transfer to open-ended reasoning tasks such as commonsense or natural-language reasoning. No experiments were provided on such domains.

4. **Missing explanation of keyword and environment generation quality.** The paper briefly states that LLMs were prompted with “keywords from BBH/KOR-Bench” but provides no justification of why these keywords cover diverse reasoning types or whether the generated tasks are semantically distinct. The role of the LLM judge is also under-specified.

**Questions:**

1. How were the ~100 keywords selected and filtered? Was this list hand-curated?
2. Did any of the generated tasks closely resemble existing datasets (e.g., GSM8K templates)?
3. How were diversity and correctness of environments quantitatively measured?

---

> ### Author Response · Authors · 2025-11-24
>
> We thank the reviewer for the detailed feedback. We will address the weaknesses and questions one by one:
>
> **Experimental details**
>
> Q: how exactly were the questions and environments generated from keywords? A: For each keyword, we prompt the LLM to come up with a task related to the keyword and implement it in the structure described in 2.2.2. We prompt the LLM n=8 times using the same keyword to get multiple related environments. Generated tasks are then filtered and revised by subsequent stages of the pipeline.
>
> Q: what proportion were filtered out, and how many survived each stage? A: Overall, around 50% of the generated tasks make it into the final dataset. We will make sure to include details about the exact number at each stage in our revisions.
>
> **Improvement from Baselines**
> - Improvement is modest: The overall performance on BBEH remains low because most subtasks are out of reach for 7B sized models. Despite this, our method achieves statistically significant improvement on 6 subtasks over the instruct model (Table 4), with particularly large gains on: causal understanding (28% -> 44%), movie recommendation (22% -> 30%), web of lies (10% -> 18%). Analysis of per task performances is provided in appendix A.3.
> - Comparison to baselines like R1-Zero: We believe R1-zero is not a direct comparison here, since it uses real math and coding data for RL training. The two closest comparisons to our method are SynLogic and Synthetic Data RL. Our advantage compared to SynLogic is that we automatically generate synthetic environments, rather than relying on a handcrafted suite, and we show stronger results in a direct comparison on reasoning benchmarks. We are aware of other works like TinyZero and Logic-RL that train on code-based synthetic environments, but they only train on a single task, so we consider SynLogic a strictly stronger baseline (in fact, the task used by TinyZero is contained in SynLogic). Our advantage compared to Synthetic Data RL is that we generate reasoning tasks implemented in code, while they try to generate ground-truth answers directly. Our ablation experiment in 5.1 indicates that code-based solutions (Code-RL and Verifier-RL) provide better supervision than raw answers (Answer-RL).
>
> **Generalizability to commonsense or natural-language reasoning**
> The BBEH and BBH benchmarks include tasks in commonsense and natural-language reasoning. In particular, we achieve large gains on causal understanding and movie recommendations (28% -> 44% and 22% -> 30%).
> Example from causal understanding (shortened for space): “Question: Lauren and Jane work for the same company. They each need to use a computer for work sometimes ... Lauren would be the only one permitted to use the computer in the mornings and that Jane would be the only one permitted to use the computer in the afternoons. ... But Jane decided to disobey the official policy. She also logged on at 9:00 am. The computer crashed immediately. Did Jane cause the computer to crash?”
>
> **Q: how were the keywords selected and filtered?**
> A: As described in section 2.2.1, there are two sources of keywords: 1. We showed an LLM example questions from BBH and KOR-bench and instructed it to identify keywords and 2. We manually added algorithm-related keywords to the list. Overall, the specific choice of BBH and KOR-bench as seed sources is not essential to our approach. They simply provide a reasonably diverse set of problem categories. Alternative methods (e.g., having an LLM propose keywords directly or manually constructing a keyword set) would also likely be viable.
>
> **Q: Did any of the generated tasks closely resemble existing datasets (e.g., GSM8K templates)?**
> A: To quantify similarity between examples, we use Claude to generate short captions for each example and embed these captions using SentenceTransformer. We find that the average similarity of GSM8K-ReSyn pairs is only 0.359, with the top 1% at 0.582. So while a small amount of overlap is possible given the simplistic structure of most GSM8K problems, the overall dataset contains mostly distinct tasks.
>
> **Q: How were diversity and correctness of environments quantitatively measured?**
> A-Correctness: The correctness of environments is implicitly checked by the difficulty calibration step described in 2.2.5 — the LLM-generated verifier must agree with generated solutions on low-difficulty instances for the task to survive. During manual inspections, we found that the rate of false positives (environments that would pass on incorrect solutions) was negligible.
> A-Diversity: To compare the diversity of our dataset with SynLogic, we conduct an analysis based on semantic clustering and included it in appendix A.5 in our updated paper.

---

### Meta-Review · Area_Chair_rudy · 2026-01-07

**Summary:**

The paper proposes ReSyn, an automated pipeline that generates synthetic reasoning environments with code-based verifiers, on which one can train reasoning LMs via reinforcement learning. Experiments on Qwen2.5-7B-Instruct show some gains on common reasoning benchmarks, suggesting the approach is promising in concept.

**Reviewer Concerns:**

Addressed: The rebuttal clarifies parts of the pipeline (e.g., filtering/keyword steps) and offers rationale for some baseline choices, plus additional analysis intended to support generalization.

Outstanding: Core reproducibility remains unresolved (missing environments/code/prompts), several baseline comparisons and statistical reporting are still absent, and concerns about validity/generalization (possible benchmark dependence/leakage; limited OOD evidence) are not convincingly closed. Several replies rely on future promises rather than concrete artifacts or results.

**Reviewer Scores:**

Despite some promising empirical improvements and an appealing idea, the work falls short on verifiability and evaluation completeness. With missing details, incomplete baselines, and weakly supported generalization claims, the reviewers do not seem convinced to recommend acceptance.

---

### Decision · Program_Chairs · 2026-01-26

Reject